



# Understanding coastal wetland conditions and futures by closing their hydrologic balance: the case of Gialova lagoon, Greece

Stefano Manzoni[1,2], Giorgos Maneas[1,3], Anna Scaini[1,2], Basil E. Psiloglou[4], Georgia
Destouni[1,2], Steve W. Lyon[1,2,5]

[1]Department of Physical Geography, Stockholm University, 10691 Stockholm, Sweden
[2]Bolin Centre for Climate Research, 10691 Stockholm, Sweden
[3]Navarino Environmental Observatory, 24001, Messinia, Greece
[4]Institute for Environmental Research & Sustainable Development, National Observatory of Athens, 15236,
Athens, Greece
[5]The Nature Conservancy, 08314 Delmont, USA

*Correspondence to*: Stefano Manzoni (stefano.manzoni@natgeo.su.se)

**Abstract.** Coastal wetlands and lagoons are under pressure due to competing demands for freshwater resources and climatic changes, which may increase salinity and cause loss of ecological functions. These pressures are particularly high in Mediterranean regions with high evaporative demand compared to precipitation. To manage such wetlands and maximize their provision of ecosystem services, their hydrologic balance must be quantified. However, multiple channels, diffuse surface water exchanges, and diverse groundwater pathways complicate the quantification of different water balance components. To overcome this difficulty, we developed a mass balance approach based on coupled water and salt balance equations to estimate currently unknown water exchange fluxes through the Gialova lagoon, SW Peloponnese, Greece. Our approach facilitates quantification of both saline and freshwater exchange fluxes, using measured precipitation, water depth and salinity, and estimated evaporation rates over a study period of two years (2016-2017). While water exchanges were dominated by evaporation and saline water inputs from the sea during the summer, precipitation and freshwater inputs were more important during the winter. About 40% and 60% of the freshwater inputs were from precipitation and lateral freshwater flows, respectively. Approximately 70% of the outputs was due to evaporation, with the remaining 30% being water flow from the lagoon to the sea. Under future drier and warmer conditions, salinity in the lagoon is expected to increase, unless freshwater inputs are enhanced by restoring hydrologic connectivity between the lagoon and the surrounding freshwater bodies. This restoration strategy would be fundamental to stabilize the current wide seasonal fluctuations in salinity and maintain ecosystem functionality, but could be challenging to implement due to expected reductions in water availability in the freshwater bodies supporting the lagoon.

## 1. Introduction

Coastal wetlands and lagoons regulate hydrologic, sediment, and contaminant exchanges between inland water bodies and the sea; they sustain biodiverse and highly productive ecosystems, and provide important ecosystem services (Thorslund et al., 2017; Newton et al., 2014). The maintenance of these essential functions is threatened by competing demands and climatic changes. Reduced freshwater inflows due to intensified water use in upstream catchments, drainage efforts to convert wetlands to agricultural areas, and construction of infrastructure that interferes with the natural water flows are all causes of reduced wetland functionality from both ecological and human perspectives (e.g., Jaramillo et al., 2018; Newton et al., 2014). In regions with high population density and





therefore higher risk of wetland overexploitation, these anthropogenic changes are compounded with ongoing
climatic trends resulting in lower precipitation and higher evaporation rates. These trends are expected to continue
and possibly worsen as global temperatures rise.

Mediterranean lagoons in particular are under pressure due to competing demands for water resources from
agriculture, tourism, fish industry and ecological systems relying on lagoons for their survival (Maneas et al.,
2019; Perez-Ruzafa et al., 2011). While runoff is already decreasing in this region (e.g., Destouni and Prieto,
2018), climatic changes are expected to increase aridity by lowering the difference between precipitation and
potential evapotranspiration by as much as 2-4 mm/day on an annual basis (Gao and Giorgi, 2008; Cheval et al.,
2017). Therefore, coastal areas in the Mediterranean region (and wetlands in particular) are regarded as vulnerable
to climatic changes (Klein et al., 2015; Gao and Giorgi, 2008).

To study the consequences of changing hydrologic regimes on wetlands (in terms of water quantity and quality),
it is necessary to quantify hydrologic fluxes. This is generally difficult due to the complex land-water interactions
in wetlands, and compound interactions of land and sea with lagoon water bodies. Lack of monitoring data for
these multiple pathways of water exchange makes the prediction of hydrologic changes challenging, requiring
indirect approaches that leverage the limited available data. To this end, overarching mass balance considerations
and closure over a whole lake or wetland system can help constrain unknown fluxes of water, solute and latent
heat (Destouni et al., 2010; Jarsjo and Destouni, 2004; Assouline, 1993). Previous examples have also
demonstrated the advantage of this approach for coastal lagoons (Martinez-Alvarez et al., 2011; Rodellas et al.,
2018). In this contribution, we develop this approach and describe a minimal coupled water-salt mass balance
model to determine both freshwater and saline flux exchanges of coastal water bodies.

While this approach is general, we focus its application on the Gialova lagoon (SW Peloponnese, Greece), the
centre of a Natura2000 protected area with a long history of water resource management that has radically altered
the lagoon functioning (Maneas et al., 2019). This hydrologic system, characterized by unquantified point and
diffuse water exchanges between lagoon and inland freshwater bodies, and between lagoon and the Ionian Sea,
offers an opportunity to apply the mass balance approach to estimate unknown hydrologic fluxes. Specifically,
we start by quantifying freshwater and saline water exchange rates over a period of two years, to evaluate the
Gialova lagoon water balance for the study period. Second, we use the results obtained for current climatic
conditions and in scenarios of altered precipitation, evaporation rate, and freshwater inputs to assess how climatic
changes and water management may alter the lagoon salinity, used here as a proxy for its ecological status.

## 2. Methods

### 2.1. Study area and climatic conditions

The Gialova lagoon is located in South West Messinia, Greece (36.97 N, 21.65 E; Fig. 1). The main water body
of the lagoon, on which this contribution focuses, covers an area of a 225.5 ha; additional 85.2 ha are covered by
surrounding wetland areas (Maneas et al., 2019). The average depth of the main water body is approximately 0.6
m (Arvanitidis et al., 1999). The lagoon is delimited by sand barriers formed around 5000 years BP
(Emmanouilidis et al., 2018), which separate the lagoon from Navarino Bay to the South and Voidokilia Bay to


the North-West. It is reasonable to assume that groundwater exchanges occur through these relatively narrow sand barriers; in addition, a permanently open canal connects Gialova lagoon to Navarino bay. The Palaiokastro hill delimits the lagoon on the West side, whereas on the North the lagoon is separated from agricultural areas by a canal built in 1960 to divert water from Xerolagados stream directly to Voidokilia bay. On the East and South-East sides, wetland areas surround the main lagoon water body. The Tyflomitis artesian springs provide freshwater

inputs in this area. Historically, the flow from these springs was approximately ≈$1.6 \times 10^6$ m$^3$ y$^{-1}$, but under current conditions it is limited to ≈$5 \times 10^5$ m$^3$ y$^{-1}$ due to water extraction upstream (Maneas et al., 2019). Furthermore, only an unknown fraction of this flow enters the wetlands and lagoon, through up-welling and surface freshwater bodies, due to a diversion canal carrying most of the spring water to Navarino Bay. On the North side, the canal draining the Xerolagados river is currently not connected with the lagoon, but groundwater from the alluvial plain

probably contributes freshwater to the lagoon at least during the wet season.

Overall, the area is characterized by Mediterranean climate, with mild wet winters and dry summers (Fig. 1). The mean annual temperature is 18 °C and the mean annual precipitation is approximately 695 mm/y (measured from 1956 to 2011 at the Helenic National Meteorological Service's station of Methoni, 15.6 km South of Gialova lagoon (Hellenic National Meteorological Service, 2019). Trends in temperature and precipitation over the

measurement period are weak and not statistically significant. The mean annual potential evapotranspiration has been estimated (Thornthwaite method) to 889 mm/y (Maneas et al., 2019).

### 2.2.    Theory

Balance equations are formulated for both water volume and salt mass in the Gialova lagoon. The lagoon receives freshwater inputs from precipitation and both surface water and groundwater fluxes, and saline water from the

Navarino and Voidokilia bays (collectively referred to as 'sea' in the following). Outputs include evaporation and water discharge to the sea. Salt is exchanged with the sea (input or output depending on flow direction), and the salt exchange fluxes depend on both the water fluxes and the salinity of the source water body. All water exchanges, except precipitation (measured) and evaporation rate (modelled based on local meteorological data), are regarded as unknown. For convenience, water and salt fluxes are defined as positive when entering the lagoon.

Water fluxes are expressed as volume per unit area of the lagoon and time (e.g., mm d$^{-1}$) and salt mass fluxes as mass per unit area and time (e.g., g m$^{-2}$ d$^{-1}$); concentrations are expressed as mass per unit volume of water (e.g., g L$^{-1}$). Subscript 'G' indicates the Gialova lagoon; subscript 'S' indicates sea water. Symbols are listed and defined in Table 1.

### 2.2.1.    Water balance equation

An overarching balance equation for water volume, neglecting water density variations, can be formulated for the lagoon in terms of its average water depth and the main water fluxes that regulate it as,

$$\frac{1}{A}\frac{dV}{dt} = \frac{dh}{dt} = P - E + Q_{fresh} + Q_{salt}, \tag{1}$$

where $A$ is the average surface area of the lagoon, $h$ is the water depth in the lagoon, $P$ and $E$ are the rates of precipitation into and evaporation from the lagoon, respectively, and $Q_{fresh}$ and $Q_{salt}$ are the exchange rates of





lateral freshwater and saline water fluxes into/from the lagoon (volumetric flow rates normalised by the lagoon area $A$). This formulation rests on the assumption that variation and potential change trend in water level are small, so that they do not alter significantly the extent of the lagoon area, which can then be sufficiently well represented by the average area $A$ over the study period. This assumption is reasonable because the shoreline is mostly limited by man-made constructions with steep walls; only at the north-west side of the lagoon a small area is seasonally flooded. Assuming essentially constant lagoon area, changes in volume ($V = hA$) can be approximated as $A$ times the changes in water depth ($\frac{dV}{dt} \approx A\frac{dh}{dt}$), justifying the first equality in Eq. (1).

In Eq. (1), precipitation rate and water depth are measured, while evaporation is estimated using Penman equation, parameterized with local meteorological data (Sect. 2.2.3). The two remaining water fluxes, $Q_{fresh}$ and $Q_{salt}$, are unknown and therefore solved for. A second equation is then necessary to obtain the two unknowns at each time step; this additional equality is provided by the salt balance.

### 2.2.2. Salt balance equation

The balance equation for the mass of salt, expressed in terms of salt mass per unit area of the lagoon reads,

$$\frac{dM}{dt} = F,$$ 

(2)

where $M$ is the salt mass per unit area (in g m$^{-2}$) and $F$ is the salt exchange rate per unit area (in g m$^{-2}$ d$^{-1}$). Following the above notations for water variables and assuming that waterborne salt transport is purely advective, the total salt mass and salt exchange fluxes can be calculated as,

$$M = C_G h,$$ 

(3)

$$F = \begin{cases} C_G Q_{salt} & Q_{salt} < 0 \\ C_S Q_{salt} & Q_{salt} > 0 \end{cases}.$$ 

(4)

In Eq. (3), the salt mass is obtained as the product of salt concentration and water depth in the Gialova lagoon ($C_G$). In Eq. (4), $Q_{salt}$ is still unknown, whereas $C_G$ is measured and the seawater salinity $C_S$ is assumed to be constant at around 38.5 g L$^{-1}$ (Civitarese et al., 2013).

Summarizing, we have now two equations in two unknowns (i.e., $Q_{fresh}$ and $Q_{salt}$):

$$\frac{dh}{dt} = P - E + Q_{fresh} + Q_{salt},$$ 

(5)

$$\frac{d(C_G h)}{dt} = \begin{cases} C_G Q_{salt} & Q_{salt} < 0 \\ C_S Q_{salt} & Q_{salt} > 0 \end{cases}.$$ 

(6)

This system of equations is solved numerically using the finite difference method (Sect. 2.2.4), yielding the unknown freshwater and saltwater exchange rates. There are uncertainties associated with almost all fluxes and compartments, but mathematically the problem is 'closed' – that is, there is enough information to obtain both $Q_{fresh}$ and $Q_{salt}$. Both mass balance equations are solved with a daily time resolution, but the results are aggregated to the monthly scale and over the whole study period.





### 2.2.3. Evaporation rate

The evaporation rate ($E$, expressed in mm/d) is calculated using Penman equation, parameterized following Duan and Bastiaanssen (2017),

$$E = \frac{1000}{\lambda \rho_w} \left[ \frac{\Delta}{\Delta + \gamma}(R_n - G) + \frac{\rho_a c_p}{\Delta + \gamma} \frac{e_s - e_a}{r_a} \right], \qquad (7)$$

where all symbols are listed in Table 1 and the factor 1000 converts units from m d$^{-1}$ to mm d$^{-1}$.

To use Eq. (7) with the available data (Sect. 2.3), several assumptions are made. First, at the daily time scale, the contribution of heat flow in the lagoon ($G$) is neglected. Second, net radiation ($R_n$) is calculated as the difference

between incoming shortwave plus longwave radiation, and outgoing shortwave plus longwave radiation, of which only incoming shortwave is measured. Reflected shortwave radiation is estimated assuming an albedo of water equal to 0.08 (McMahon et al., 2013). Net outgoing longwave radiation is estimated using an empirical relation that accounts for both surface temperature and atmospheric conditions that affect incoming longwave radiation (Allen et al., 1998). In this relation, increasing vapour pressure and decreasing solar radiation (both quantities are

measured at our site) decrease outgoing longwave radiation for a given surface temperature. Third, the aerodynamic resistance is parameterized for open water evaporation following Shuttleworth (2012). To test this parameterization, the ratio of equilibrium evaporation and total evaporation was computed, resulting in a median value of 1.35 (first and third quartiles: 1.21 and 1.62, respectively). While this median value is higher than the classical result by Priestley and Taylor (1972) for wet vegetated surfaces (i.e., 1.26), our values are well within

the range estimated for waterbodies (Assouline et al., 2016). This result thus lends support to the adopted parameterization. Finally, and similar to other studies using the Penman approach to estimate evaporation rate (e.g., Rosenberry et al., 2007; Martinez-Alvarez et al., 2011; Rodellas et al., 2018), we assume that conditions over the lagoon are homogeneous, and that our point measurements are representative. Considering the relatively small size of the lagoon, this assumption is deemed reasonable.

### 2.2.4. Numerical approach to solve the balance equations

The known quantities in Eq. (5) and (6) include $P$, $C_G$, and changes in water depth ($\frac{dh}{dt} \approx \frac{\Delta h}{\Delta t}$ over a fixed, not infinitesimal, time interval), and $E$ is estimated as described in Sect. 2.2.3. To proceed, $Q_{fresh}$ and $Q_{salt}$ in Eq. (5) and (6) must be expressed as functions of these known quantities. Eq. (6) allows finding $Q_{salt}$ as,

$$Q_{salt} = \begin{cases} \frac{1}{C_G} \frac{d(C_G h)}{dt} & Q_{salt} < 0 \\ \frac{1}{C_S} \frac{d(C_G h)}{dt} & Q_{salt} > 0 \end{cases}. \qquad (8)$$

Using the chain rule of differentiation and discretizing through time we obtain,

$$\frac{d(C_G h)}{dt} = h \frac{dC_G}{dt} + C_G \frac{dh}{dt} \approx h \frac{\Delta C_G}{\Delta t} + C_G \frac{\Delta h}{\Delta t}, \qquad (9)$$





where $\Delta t$ is the discretization time step. If $h\frac{\Delta C_G}{\Delta t} + C_G\frac{\Delta h}{\Delta t} > 0$, salt mass increases and it follows that $Q_{salt} > 0$, so

that the second expression in Eq. (8) is used to obtain $Q_{salt}$. In contrast, if $h\frac{\Delta C_G}{\Delta t} + C_G\frac{\Delta h}{\Delta t} < 0$, salt mass decreases

and the first equation is used. Therefore, combining Eq. (8) and (9) we finally obtain,

$$Q_{salt} = \begin{cases} \frac{1}{C_G}\left(h\frac{\Delta C_G}{\Delta t} + C_G\frac{\Delta h}{\Delta t}\right), & h\frac{\Delta C_G}{\Delta t} + C_G\frac{\Delta h}{\Delta t} < 0 \\ \frac{1}{C_S}\left(h\frac{\Delta C_G}{\Delta t} + C_G\frac{\Delta h}{\Delta t}\right), & h\frac{\Delta C_G}{\Delta t} + C_G\frac{\Delta h}{\Delta t} > 0 \end{cases} . \tag{10}$$

The next step requires discretizing and solving Eq. (5) to obtain $Q_{fresh}$ from the other hydrologic fluxes, changes

in water depth, and $Q_{salt}$ from Eq. (10),

$$Q_{fresh} = \frac{dh}{dt} - P + E - Q_{salt} \approx \frac{\Delta h}{\Delta t} - P + E - Q_{salt}, \tag{11}$$

The two linked Eq. (10) and (11) do not need to be coupled through time because changes in water depth and salt

concentration in the lagoon are measured. These equations could thus be solved for each time interval in sequence.

Since the absolute value of $h$ varies from one time step to the next, and this absolute value is not measured, $h$ in

Eq. (10) must be updated at each time step as $h_{t+1} = h_t + \Delta h$ before being entered in the equation for the next

time step. As long as $\Delta h/h \ll 1$, this step would not be necessary, but since water depth fluctuations can be

significant with respect to the mean depth in the shallow Gialova lagoon, this correction is important.

To summarize, Eq. (10) and (11) represent a simple algorithm to calculate the unknown exchanges of water

between Gialova lagoon and the freshwater systems upstream and the sea downstream. Moreover, they provide

estimates of salt mass fluxes associated with the saline water exchanges.

### 2.2.5.   Simulation scenarios

To assess the effects of changing climatic conditions and water resource management on salinity in the Gialova

lagoon, we solved Eq. (5) and (6) in a forward mode – that is, to estimate salinity variations through time based

on known hydrologic fluxes. Measured precipitation and evaporation rates are modified to account for climatic

changes, $Q_{fresh}$ is modified to account for both climatic and water management changes, and the change in storage

($dh/dt$) is maintained from current conditions given the strong coupling of water levels in the lagoon and in the

sea (Supplementary materials S1; Fig. S1). Sea level rise is not considered in these scenarios. Based on these

water fluxes and storage changes, $Q_{salt}$ is calculated using the water balance Eq. (5), and salinity is obtained with

the salt mass balance Eq. (6).

We considered three climate change scenarios (Table 2): C1) reduced precipitation, C2) increased temperature

(and thus evaporation), and C3) combined reduction in precipitation and increase in temperature. All scenarios

are based on results of a regional climate model forced by global climatic conditions under high $CO_2$ emissions

(denoted as A2), with predictions extending to year 2100 (Gao and Giorgi, 2008). For C1, a precipitation reduction

up to 30% was considered (while keeping current $E$); for C2, evaporation was increased up to 20% due to a mean

annual temperature increase of 4°C (while keeping current $P$); for C3, the changes in $P$ and $E$ were compounded.

Moreover, to assess the effects of changes in water-atmosphere exchanges in isolation, the lateral freshwater fluxes





were either maintained as under current conditions, or decreased to represent reductions in runoff for drier conditions driven by climate change.

In the simulations where lateral freshwater fluxes are decreased, a reduction coefficient was applied to all daily values of freshwater exchanges (water entering and leaving the lagoon). The reduction coefficients for each climate scenario were obtained by estimating future runoff $R$ from catchments surrounding the lagoon using

Budyko's approach (Choudhury, 1999). First, the relation between long-term actual evapotranspiration (AET) and potential evapotranspiration (PET) was parameterized following Choudhury (1999). Second, long-term runoff was calculated as $R=P$-AET under the assumption of negligible change in water storage in the catchment,

$$R = P - \text{AET} = P - \frac{P\,\text{PET}}{(P^n + \text{PET}^n)^{\frac{1}{n}}}, \tag{12}$$

where $n$=1.8 (Choudhury, 1999). Assuming PET$\approx E$, Eq. (12) allows estimating $R$ when $P$ and PET change according to the climate scenarios C1-C3. Finally, reduction coefficients for the freshwater exchanges are

calculated as the ratios of $R$ under future conditions (from Eq. (12)) over $R$ under current conditions (i.e., $R$=167 mm y$^{-1}$ for $P$=695 mm y$^{-1}$ and PET=889 mm y$^{-1}$, see Sect. 2.1). The obtained reduction coefficients are reported in Table 2.

These climatic scenarios were further combined with altered water resource management scenarios, in which we assumed that the lateral freshwater fluxes are either reduced due to intensified water use, or increased by attempts

to restore Gialova lagoon to its original state of a brackish wetland (Table 2). To consider a wide range of possible management outcomes, we considered freshwater flux changes varying continuously from a 50% reduction to a 50% increase with respect to the current conditions, as estimated in Sect. 2.2.4. Management scenarios were implemented by varying the daily modelled values of $Q_{fresh}$, independently of climatic conditions (e.g., assuming that freshwater exchanges are fully controlled and not limited by water availability). As also done for the climate

change scenarios, all daily values of $Q_{fresh}$ were varied, including freshwater losses from the lagoon, thereby imposing a control on the two-way connectivity of the lagoon with the surrounding freshwater bodies.

### 2.3.    Measurements

#### 2.3.1.    Permanent equipment

Meteorological measurements were conducted with a Decagon Devices, Inc. system, including a relative humidity

and air temperature sensor (VP4), a 2-D sonic anemometer (DS-2), a solar radiation sensor (pyranometer), and a rain gauge (ECRN-100), all installed in March 2016. Data were recorded using EM-50 loggers. These pieces of equipment, except for the anemometer, were located on the Southern shore of the lagoon; the anemometer was installed on a concrete pillar in the middle of the lagoon (approximately 1 km from the other sensors) to minimize interference by vegetation and nearby terrain. Three water quality/depth measurement points were set up in the

lagoon – one at the Southern shore next to the meteorological station, one next to the concrete pillar, and one on a PVC pole located near the Northern shore (Fig. 1a). The Southern and Northern measurement sites were equipped with one conductivity-temperature-depth probe, while the central site had two probes at different depths (all CTD-10). For this study, salinity was estimated from the average value of the two central sensors, considered

as representative for the whole lagoon (Supplementary materials S2; Fig. S2). Missing data from late 2017 and

January 2018 were gap-filled using the time series from the Northern measurement site, which is well-correlated with that from the central measurement site (Fig. S3). Water depth variations were obtained from the Northern sensor, which had no missing values and had remained at the same vertical position throughout the duration of the monitoring. The CTD-10 at the Southern shore was not used as it exhibited significant tidal fluctuations in salinity due to the nearby channel connecting Gialova lagoon and Navarino bay, and thus cannot be regarded as

representative for the whole lagoon.

### 2.3.2. Field campaigns and areal estimates of salinity

The representativeness of the salinity measurements from the central site was tested by comparing the point measurements to distributed measurements collected along the shore and in the lagoon during several intensive campaigns from 1995 to 2018 (Supplementary materials S2). The campaigns were conducted at different times of

the year to span the full range of salinities occurring in the lagoon. Distributed data from these campaigns were used to estimate areal average salinities that were compared to the point measurements described in Sect. 2.3.1. The comparison yielded a linear relation close to the 1:1 line ($R^2$=0.95) that was used to scale up the point measurements to the whole lagoon area (Fig. S2). A summary of the areal salinity estimates from all the measurement campaigns is also shown in Fig. S4 as a function of month to illustrate its seasonal cycle.

### 2.3.3. Data processing

Meteorological data were collected with a sampling interval of five minutes and averaged (or accumulated for rainfall and solar radiation) over a day. Gaps in the meteorological data record were filled using time series from two other meteorological stations – one located in an olive orchard 5 km from Gialova lagoon, the other located at Methoni (National Observatory of Athens, 2019). Details on the gap filling procedure are reported in the

Supplementary materials S3.

Water electrical conductivity measurements included erroneous readings (downward peaks with unreasonably low conductivity values with respect to a well-defined upper envelope). These erroneous values were removed with a two-step de-spiking algorithm before daily averaging. The algorithm first detected and removed outliers (values lower than the 10th percentile) in a moving window of four hours. Second, data points that caused the

standard deviation in the moving window to be higher than four times the standard deviation in a window without any error were also removed. Conductivity values (expressed in mS cm$^{-1}$) were converted to salt concentrations (g L$^{-1}$) using the empirical relation, $C_G = 0.4665 EC^{1.0878}$, where $EC$ is the electrical conductivity at 25 °C (Williams, 1986) (temperature corrections were performed when logging the data).

Salinity in the Ionian Sea is set to a constant value of 38.5 g L$^{-1}$. This value was obtained from one meter depth,

monthly data for the period 2008-2012 from the measurement buoy number 68422 (Pylos) of the European Marine Observation and Data Network (French Research Institute for Exploitation of the Sea, 2018). The buoy was located approximately 10 km South West of Gialova lagoon and is assumed representative of the salinity in both Navarino and Voidokilia bays. Seasonal variations in salinity were minor (38.2 to 38.9 g L$^{-1}$) compared to the



variability of salinity in the Gialova lagoon, thereby supporting our assumption of time-invariant sea water

salinity.

## 3. Results

### 3.1. Water fluxes under current conditions

The study period is characterized by typical Mediterranean conditions, with mild and wet winters, and warm and dry summers (Fig. 2-3). Even though winters receive most of the rainfall, some late summer storms were

exceptionally intense (140 mm between DOY 250 and 251 of 2016). This precipitation regime, together with intense summer evaporation rates sustained by high solar radiation and temperature (Fig. 3d), contributes to strong seasonal variations in salinity (Fig. 2d). Salinity tends to increase during the spring and throughout the summer, peaking in late summer or early fall (hypersaline conditions). Fall and winter freshwater inputs eventually restore salinity values below values typical of the Ionian Sea. These seasonal fluctuations are consistent with those

reported in earlier studies (summarized in Fig. S4) and depend on both hydro-climatic conditions (both intra- and inter-annual fluctuations in water balance) and inter-annual changes in water management (Sect. 4.3).

The seasonal pattern of salinity is punctuated by sudden events in response to intense rainfall events. The most notable event occurred on DOY 251 of 2016, leading to a rapid decrease in salinity from 35 to 22 g L$^{-1}$, with the lower lever sustained over the following month and a half. High salinity is associated with high water levels (Fig.

2c), which tend to occur during the warmer months despite their higher evaporation rate. Water levels vary largely as a function of tidal fluctuations in the Ionian Sea (Supplementary materials S1), and are poorly related to individual rainfall events or the seasonal fluctuations in hydro-climatic variables. Specifically, the water levels in the Gialova lagoon lag approximately one day behind those of the Ionian Sea (Pearson correlation coefficient is maximized with a lag of one day, $r$=0.78; Fig. S1).

Fig. 4a illustrates monthly averaged hydrologic fluxes, including precipitation, evaporation and changes in water depth in the lagoon. While evaporation is generally higher than precipitation, as expected in Mediterranean climates (Fig. 1), variations in storage (i.e., water depth) are more dynamic than in the other two lagoon-atmosphere fluxes, and do not compensate for the negative hydrologic balance of the lagoon. Given the necessity of water balance closure over the lagoon, this suggests that other water exchanges via groundwater and surface

water play a significant role. The coupled water and salt mass balances (Eq. (10) and (11)) facilitate estimation of the freshwater and saline water exchanges between the Gialova lagoon and surrounding water bodies (Fig. 4b), through closure of the lagoon hydrologic balance. The estimated fluxes are in the same order of magnitude of rainfall and evaporation rates, reaching monthly averages of ±10 mm d$^{-1}$, and highly variable. Freshwater exchanges are generally positive, indicating inputs to the lagoon, as might be expected by the presence of a

freshwater aquifer, as well as diffuse and point inputs from streams and springs feeding the main water body of the lagoon. However, during the sudden salinity increase on DOY 294 of 2016, the model suggests a rapid outflow of freshwater (where outflow is a negative flux). This outflow is consistent with the constraint set by the salt water balance and is needed to explain the measured increase in salt concentration at nearly constant water level. Throughout the study period, the saline fluxes often change sign, indicating alternate periods during which either





water from the Ionian Sea enters the lagoon (positive sign, generally during the summer months), or water from the lagoon flows into the sea (negative sign, generally during winter and spring).

Some of the daily fluxes shown in Fig. 2 and 3, and the calculated saline and fresh water exchange fluxes are correlated (Table 3). Precipitation and evaporation rates are negatively correlated due to their seasonal cycle (Fig. 2a and 3d), and variations in water level are positively correlated to precipitation rates. These correlations emerge

from intrinsic processes and relations among hydrologic variables that are not explicitly parameterized in the present water and mass balance equations, whereas other correlations are expected from the physical balance relations expressed mathematically in Eq. (10) and (11). In particular, saline water fluxes are (strongly) positively correlated with water level variations and (strongly) negatively correlated with freshwater fluxes, as implied by Eq. (11). The occurrence of the strong correlation between fresh and saline water fluxes indicates that variability

in water exchanges between the lagoon and the sea or land dominates over variability imposed by water-atmosphere exchanges.

When considering the entire study period, precipitation represents ≈40% of the water inputs to the lagoon, whereas the remaining 60% is driven by freshwater exchanges, with a minor contribution by change in water level (Fig. 5a, left bar). Evaporation represents ≈70% of the water outputs from the lagoon, and the remaining 30% is caused

by saline water loss from the lagoon to the Ionian Sea (Fig. 5a, right bar).

### 3.2. Water fluxes under changes in climate and water management

Climatic changes reducing precipitation and increasing evaporation rate (scenario C3) are expected to alter the water balance with respect to current conditions (Fig. 5b). The overall lower inputs and higher evaporative losses are compensated by lower outputs of saline water from the lagoon. Note that in Fig. 5b we assumed that freshwater

inputs remained as today, but without any management effort, the flows from the surface freshwater bodies would most likely decrease as a result of lower precipitation and higher evapotranspiration in the catchments feeding the lagoon (Sect. 4.3). Fig. 6a shows predicted changes in the mean salinity of Gialova lagoon as a function of gradual precipitation reduction on the abscissa, in combination with unchanged (scenario C1) or increased evaporation rate (C3) while keeping current freshwater inputs (black lines). In addition, the effect of reduced freshwater inputs

due to lower runoff from the surrounding catchments is considered (red lines). In all cases, salt concentrations increase with decreasing precipitation and increasing evaporation rate due to accumulation of salt from the marine sources. Salt accumulates because it enters the lagoon during increases in sea water level, but it does not leave it again under decreases in sea level because the saline outflows are then also decreased (Fig. 5b). However, when both precipitation and freshwater inputs are decreased, salinity increases more than when only precipitation is

decreased (compare red and black lines).

The time spent under hypersaline conditions is a more useful ecological indicator than mean salt concentration, as it directly impacts the ecological actors. Fig. 7a shows how the percentage of time under hypersaline conditions varies with hydro-climatic changes (similar to those in Fig. 6a). Decreasing precipitation alone increases moderately the time in hypersaline conditions from the current 3 months per year. As noted for changes in salinity

(Fig. 6a), higher evaporation rates – especially when compounded with lower freshwater inputs – yield longer periods under hypersaline conditions, potentially up to most of the year in the worst-case scenario.


Altering the management of surface freshwater also affects salinity in the Gialova lagoon, as shown in Fig. 6b. Allowing more (less) freshwater to flow into the lagoon leads to decrease (increase) in salinity under any climate scenario. The change in salinity is nonlinearly related to changes in freshwater input, consistent with an overall

salt dilution effect. Under current climatic conditions (dotted line in Fig. 6b), a 50% reduction in freshwater inflows leads to an annual mean salinity similar to that of the Ionian Sea. Under future climatic conditions, a similar decrease in freshwater inputs increases salinity in the lagoon to values that are much higher than in the Ionian Sea. To prevent a transition to a saline lagoon under future climatic conditions, freshwater inputs need to be increased by ≈25, 30 or more than 50%, respectively, under the three scenarios of increased aridity (C1, C2,

and C3, respectively). These percentages can be seen in Fig. 6b at the intersections between the current salt concentration (horizontal grey dotted line) and the three curves representing salt concentrations under the three climate scenarios.

Similar patterns emerge in Fig. 7b for the duration of hypersaline conditions. With lower freshwater inputs combined with climatic changes, the model predicts hypersaline conditions for up to 9 months per year. Only

increasing freshwater inputs, in absence of climatic changes, limits hypersaline conditions to about 2.5 months per year.

## 4. Discussion

### 4.1. Approach limitations

The proposed approach rests on several assumptions that may affect our results and conclusions. First, this is a

lumped approach that neglects spatial variability by assuming well-mixed conditions vertically and laterally. We assume that salinity in the central measurement point (averaging measurements at two depths) is representative of average salinity over the whole lagoon. While this might not be always the case, especially after strong rain events causing high localized freshwater inputs from the sluice gate connecting the Tyflomitis diversion canal to the lagoon (Maneas et al., 2019), we have shown that variability in spatial average salinity is well captured by the

point measurements (Fig. S2). However, salt fluxes depend on salinity levels where hydrological flows occur. Thus, a lumped approach has the potential to overestimate salt fluxes by over-emphasizing salinity gradients. Assessing the consequences of this assumption requires a detailed hydrodynamic model of the lagoon. However, our lumped approach is still comparable in scope to previous efforts on lakes and lagoons where no distributed salinity data are available (Assouline, 1993; Martinez-Alvarez et al., 2011; Rodellas et al., 2018).

The water balance Eq. (1) constraints estimates of the unknown freshwater inputs, implying that any uncertainty in the other water fluxes affects the freshwater input results. For example, a sudden increase in salinity not associated with significant water level change leads to both saline water inputs into and freshwater losses from the lagoon, in order to close both the salt and the water balance. The freshwater losses are difficult to interpret, but could be explained as loss of relatively fresh water far from the lagoon-sea canal, which is substituted by more

saline water from the Ionian Sea. It would be valuable to monitor two-way water exchanges at the sluice gates by regulating water exchanges from inland water bodies to test this model implication. In the long-term, however, freshwater inputs dominate the water balance, as expected under a Mediterranean climate with relatively low precipitation.


Moreover, for our well-mixed assumption to hold, the used water and salt balances cannot be resolved at time scales shorter than the equilibration time for the lagoon, which we estimate to be on the order of one day (Fig. S1). Therefore, significant exchanges of water and salt occurring due to shorter term fluctuations in sea water level are neglected but could be important at longer time scales. To address the limitations of the well-mixed assumption and gain insights on water exchanges at sub-daily time scales, the Gialova lagoon should be

represented by a spatially explicit hydrodynamic modelling approach; such approaches have been developed and used in other recent studies for simulation of coastal and semi-enclosed sea conditions at various scales and under different hydro-climatic and/or water management scenarios (Chen et al., 2019; Vigouroux et al., 2019).

Another source of uncertainty may be introduced by attributing changes in electrical conductivity solely to salinity changes (Williams, 1986). Other solutes can impact electrical conductivity but not salinity, potentially leading to errors. For the current application, where we focus on relatively short time periods, there are likely minimal land

cover or nutrient load changes, which limits the variability of electrical conductivity due to other constituents. Also, assuming relatively stable flow pathway distributions in the landscape over the study period, the geochemical signature of groundwater is likely constant enough so that the main driver of changes in electrical conductivity is salinity from sea water rather than terrestrial sources.

### 4.2. The hydrologic balance of the Gialova lagoon under current and future climate

In the absence of direct flow measurements, the presented water balance approach provides estimates of the major hydrologic fluxes exchanged through the Gialova lagoon (Fig. 4), allowing to assess its overall water balance over the two-year study period. We estimate large two-way exchanges with the Ionian Sea, facilitated by the canal that connects the lagoon with Navarino Bay (minimum cross section of 7 m$^2$). Saline water inflows amount to 2490 mm y$^{-1}$ and outflows to 3200 mm y$^{-1}$, with a net loss of water from the lagoon to the sea of 710 mm y$^{-1}$ (with all

flows normalised by the average surface area of the lagoon). These gross water fluxes are comparable to those exchanged through Mar Menor – a much larger lagoon in southern Spain, connected to the Mediterranean sea via five channels (Martinez-Alvarez et al., 2011) – whereas the volumetric flows exchanged by the Gialova lagoon are lower due its smaller size. The seasonal pattern characterized by freshwater inputs in the winter and spring and salt water outputs primarily in the summer and autumn is similar to that observed in other Mediterranean

lagoons (Stumpp et al., 2014). The estimated freshwater inputs amount to 1170 mm y$^{-1}$, which are mainly partitioned between evaporation (≈40%) and water losses to the Ionian Sea (≈60%). Currently, the main freshwater input to the lagoon is from Tyflomitis stream and artesian springs. Flow estimates for these water sources are uncertain and currently range between 0.5×10$^6$ and 2.0×10$^6$ m$^3$ y$^{-1}$ (Maneas et al., 2019), equivalent to approximately 220 to 890 mm y$^{-1}$ of freshwater inputs after normalizing the flow rates by the lagoon area. Our

estimated freshwater inputs are higher but still reasonable, considering that part of the water from Tyflomitis is diverted to Navarino bay, but that unquantified groundwater flows likely contribute freshwater to the lagoon as well.

The Mediterranean region is projected to become warmer and drier (Cheval et al., 2017; Gao and Giorgi, 2008). By reducing precipitation and associated runoff, and increasing evaporation rates, future climatic conditions are

expected to increase salinity unless freshwater inputs now diverted from around the lagoon to the Ionian Sea are restored. To assess the consequences of these climatic changes, we first calculated how salinity changes in

response to independent variations in precipitation, evaporation, and freshwater inputs (black curves in Fig. 6). Results show – as expected – increases in salt concentrations for all these hydrologic changes. Similarly, the time spent under hypersaline conditions is expected to increase with such climatic changes and without intervention

measures (black curves in Fig. 7). However, it is reasonable to expect a reduction in the freshwater inputs due to both lower runoff from surrounding catchments under more arid conditions (Eq. (12)), and increased water abstractions (Sect. 4.3). Using the Budyko curve to estimate changes in runoff from the catchments feeding the Gialova lagoon (Eq. (12)), we found that a reduction in precipitation, an increase in PET, and both changes together (as in Table 2), respectively cause runoff and thus freshwater inputs to decrease by 57, 21, and 67% from

the current level. These reductions worsen the lagoon conditions, increasing salinity during winter and spring more than predicted when climate does not constrain freshwater inputs.

We have not assessed the effects of long-term changes in sea level. The water levels in the Gialova lagoon are well-coupled to those in the Ionian Sea, with a time lag of one day (Fig. S1). As a result of this tight connection, sea level rise is expected to be mirrored by a rise in lagoon water level. This may alter the exchanges of saline

water with Navarino Bay, since the low Divari sand barrier may be eroded, creating more channels that link the lagoon to the sea. Faster exchanges with Navarino Bay imply that hypersaline conditions during the summer may be avoided, but also that the duration of periods with brackish water would then be shortened. Resolving this question is beyond the scope of this study, but can be addressed in future studies coupling projections of sea level rise to possible geomorphological changes in the area.

**4.3.**      **Management options for salinity regulation**

Water and land in the Peloponnese peninsula, where Gialova lagoon is situated (Fig. 1a), have been intensively used at least for the last 6000 years, often resulting in major changes in hydrologic functioning (Butzer, 2005; Weiberg et al., 2016). Land conversion to agriculture during periods of population growth and back to natural vegetation when pressure decreased, resulted in periods of sustained erosion, change in soil surface properties,

altered evapotranspiration rates associated with varying vegetation types, and modifications of surface flow pathways due to construction of terraces and other water-conserving systems. These alterations have likely caused variable freshwater inputs to coastal lagoons, which have been compounded with long-term hydro-climatic trends and geomorphological processes (Emmanouilidis et al., 2018). Arguably, the most intense alterations occurred after 1960, when the streams feeding the lagoon were diverted, and the lagoon was drained and isolated from the

sea to expand agricultural areas (Maneas et al., 2019). Since 1976, however, hydrologic linkages with the sea and partly with the surface freshwater sources have been restored. In 1999, sluice gates were also opened to connect the drainage canals to the lagoon. Despite these restoration efforts, today the Gialova lagoon remains poorly connected with freshwater sources, and is thus more saline than before the drainage (Fig. 2 and S4). Earlier measurements support this view: the opening of the sluice gates increased surface hydrological connectivity and

allowed salinity to decrease from summer values above 60 g L$^{-1}$ to between 30 and 50 g L$^{-1}$ (compare data from the 1995-1996 field campaign to those of the later campaigns in Fig. S4).

As also shown in previous studies (Arvanitidis et al., 1999; Koutsoubas et al., 2000), salinity varies seasonally, with higher values during the dry period (June – October) and lower values during the wet period (November – March) (Fig. S4). However, in contrast to these studies our findings indicate that water conditions should be



characterized as saline or/and even hypersaline during the months June – November, with more brackish
conditions occurring only during the wetter winter and spring seasons. High salinity has been highlighted in
previous studies as one of the main factors leading to dystrophic crisis events and fish mortality (Arvanitidis et
al., 1999; Koutsoubas et al., 2000). The combined effects of increased salinity and limitation in water circulation
has led to extensive reed and cattail mortality, thereby deteriorating important habitats for water birds (Maneas et

al., 2019). On one hand, the lagoon has an important fishery value (Koutsoubas et al., 2000), and the opening of
additional canals with the Navarino bay has been suggested as a way to improve water circulation and fishing
(Arvanitidis et al., 1999). On the other hand, the lagoon is part of a wider protected bird area. Increased freshwater
inputs could be favourable for bird conservation, but could also negatively affect the fishing in the lagoon if
freshwater inputs are of low quality due to agricultural contaminants. A sustainable future water management

strategy should aim to create favourable conditions for both fishing and bird conservation. Such a strategy will
need to be based on more data, from not only systematic monitoring of fish stocks, bird status and water parameters
such as oxygen levels and nutrients concentrations, but also from knowledge from local fishermen.

In the current context of water scarcity and competing water demands in the Mediterranean region – and Greece
in particular (Destouni and Prieto, 2018; Klein et al., 2015; Perez-Ruzafa et al., 2011) – managing freshwater

inputs to the Gialova lagoon can be challenging. Our results show that, to adapt to expected climatic conditions
by the end of 2100 and maintain the current annual average salinity in the lagoon, a more than 50% increase in
freshwater inputs should be achieved, corresponding to at least 1750 mm $y^{-1}$ (Fig. 6b). To assess if such a 50%
increase is feasible, the total water resources currently available must be quantified. Combining the flow rates of
the two streams that before being diverted reached and fed the lagoon (Maneas et al., 2019), the total surface water

that would be naturally available is between 2400 and 3100 mm $y^{-1}$ (values are normalized by lagoon area and
include the contributions from Tyflomitis springs reported in Sect. 4.2), which in principle could fulfil the demand
to stabilize salinity. These estimates are higher than, but considering the uncertainties still comparable with the
2200 mm $y^{-1}$ obtained from the Budyko curve (Eq. (12)). These estimates do not reflect the expected reduction in
precipitation and increase evaporation in the coming decades. In the worst-case climate scenario of 67% lower

runoff (Sect. 4.2), the available surface water inflow could be limited to the range 810-1030 mm $y^{-1}$ – clearly not
enough to increase the current estimated freshwater inputs. This estimate is conservative; if water abstraction
increases as might be required to maintain agricultural productivity in the catchment (Maneas et al., 2019),
freshwater inputs would be even lower. Thus, under future conditions, managing freshwater inputs to maintain
current salinity levels appears unlikely.

**5.   Conclusions**

We have shown through a mass-balance approach that under current climatic conditions the Gialova lagoon
receives about 40% of water inputs from precipitation and 60% from surface and groundwater freshwater sources.
Under these conditions, the lagoon is hypersaline for nearly 30% of the year. Under future, drier and warmer
conditions, the water balance is predicted to change towards higher water losses, associated with higher salinity

levels and in the worst-case scenario prevalence of hypersaline conditions. The same modelling approach suggests
that managing the freshwater inputs available today could reduce salinity. However, under future conditions such



efforts might not be enough to stabilize salinity to current levels because runoff will likely be reduced by climatic factors and possibly higher water abstractions.

**Data availability**

Should the manuscript be accepted, the primary dataset (meteorological and water quality parameters) will be permanently archived in the open-access database of the Bolin Centre for Climate Research (preliminary webpage of the dataset: https://bolin.su.se/data/manzoni-2019). Note that this database is listed in https://www.re3data.org/ as required.

**Author contribution**

SM and SWL developed the mathematical model with feedback from GD; SM and GM collected and analysed the 2016-2018 data; GM collected and analysed the historical data; AS conducted the spatial analysis in Supplementary Sect. S2; BEP provided and analysed meteorological data from the NOA/IERSD station at Methoni; SM drafted the manuscript; all authors discussed the content, provided feedback and wrote the final manuscript.

**Competing interests**

The authors declare that they have no conflict of interest.

**Acknowledgements**

We thank MSc students Agnes Classon and Kim Lundmark for assistance installing the equipment and preliminary data analyses, and Eirini Makopoulou for preparing Fig. 1a. The authors acknowledge support from
the Navarino Environmental Observatory (NEO), a partnership between Stockholm University, the Biomedical Research Foundation of the Academy of Athens (BRFAA) and TEMES S.A. GM and GD acknowledge the EU Horizon 2020 project COASTAL (H2020-RUR-02-2017, grant 773901). AS and SM were partly supported by the Bolin Centre for Climate Research, and by the Swedish Research Councils (Vetenskapsrådet, Formas) and Sida through the joint project VR 2016-06313. This study has been conducted using sea water salinity data from
E.U. Copernicus Marine Service Information, and sea water levels from the UNESCO Intergovernmental Oceanographic Commission.





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


**Table 1. Definition of symbols used in the mass balance model and in the calculation of the evapotranspiration rate.**

| Symbol | Description | Units |
|---|---|---|
| | Water and salt balance model (Eq. (1)-(6)) | |
| $A$ | Gialova lagoon area | $m^2$ |
| $C_G$ | Salt concentration in the Gialova lagoon | $g\ L^{-1}$ |
| $C_S$ | Salt concentration in the Ionian Sea (=38.5 g L$^{-1}$) | $g\ L^{-1}$ |
| $E$ | Evaporation rate | $mm\ d^{-1}$ |
| $F$ | Salt mass exchange rate | $g\ m^{-2}\ d^{-1}$ |
| $h$ | Gialova lagoon water depth | mm |
| $M$ | Salt mass per unit area in the Gialova lagoon | $g\ m^{-2}$ |
| $P$ | Precipitation rate | $mm\ d^{-1}$ |
| $Q_{fresh}$ | Freshwater exchange rate (including surface and groundwater) | $mm\ d^{-1}$ |
| $Q_{salt}$ | Saltwater exchange rate (including surface and groundwater) | $mm\ d^{-1}$ |
| $V$ | Water volume in the Gialova lagoon | $m^3$ |
| | Runoff model (Eq. (12)) | |
| AET | Actual evapotranspiration | $mm\ y^{-1}$ |
| PET | Potential evapotranspiration | $mm\ y^{-1}$ |
| $R$ | Runoff from catchments surrounding the Gialova lagoon | $mm\ y^{-1}$ |
| | Evaporation model (Eq. (7)) | |
| $c_p$ | Heat capacity of air | $MJ\ kg^{-1}\ ^oC^{-1}$ |
| $e_s$ | Saturated atmospheric vapour pressure | kPa |
| $e_a$ | Actual atmospheric vapour pressure | kPa |
| $G$ | Heat flow in the water column | $MJ\ m^{-2}\ d^{-1}$ |
| $r_a$ | Aerodynamic resistance | $d\ m^{-1}$ |
| $R_n$ | Net radiation | $MJ\ m^{-2}\ d^{-1}$ |
| $\varDelta$ | Slope of the vapour pressure saturation-temperature relation | $kPa\ ^oC^{-1}$ |
| $\gamma$ | Psychrometric constant | $kPa\ ^oC^{-1}$ |
| $\lambda$ | Latent heat of vaporization | $MJ\ kg^{-1}$ |
| $\rho_a$ | Air density | $kg\ m^{-3}$ |
| $\rho_w$ | Water density | $kg\ m^{-3}$ |





**Table 2. Climatic and management scenarios. Variations are indicated as percentage change compared to**
**current conditions. In scenarios C1-C3, freshwater inputs are either kept as under current conditions or**
**decreased as a consequence of higher potential evapotranspiration/precipitation ratio (respectively black**
**and red curves in Fig. 6a and 7a).**

| Scenario | Code | Explanation | Change in precipitation $P$ (% of current) | Change in evaporation rate $E$ (% of current) | Change in freshwater input $Q_{fresh}$ (% of current) | Source for $P$ and $E$ variations |
|---|---|---|---|---|---|---|
| Climate (Fig. 6a, 7a) | C0 | Current conditions | 0% | 0% | 0% | Fig. 4 |
| | C1 | Reduced $P$ | 0 to -30% | 0% | 0% to -57% | (Gao and Giorgi, 2008) |
| | C2 | Increased $E$ | 0% | 0 to +20% | 0% to -21% | (Gao and Giorgi, 2008) |
| | C3 | Reduced $P$, increased $E$ | 0 to -30% | 0 to +20% | 0% to -67% | (Gao and Giorgi, 2008) |
| Management and climate (Fig. 6b, 7b) | Decreased or increased $Q_{fresh}$ | | All climatic scenarios C0-C3 | All climatic scenarios C0-C3 | -50% to +50% (independent of $P$ and $E$) | Assumed |





**Table 3. Pearson correlation coefficients between pairs of hydrologic fluxes (or change in storage), as calculated with Eq. (10) and (11) at the daily time scale (\* indicate significant correlations, p<0.001).**

|  | $dh/dt$ | $P$ | $E$ | $Q_{fresh}$ | $Q_{salt}$ |
|---|---|---|---|---|---|
| $dh/dt$ | 1.00 | 0.29* | 0.00 | 0.05 | 0.63* |
| $P$ | | 1.00 | -0.29* | -0.14* | -0.05 |
| $E$ | | | 1.00 | 0.05 | 0.15* |
| $Q_{fresh}$ | | | | 1.00 | -0.66* |
| $Q_{salt}$ | | | | | 1.00 |

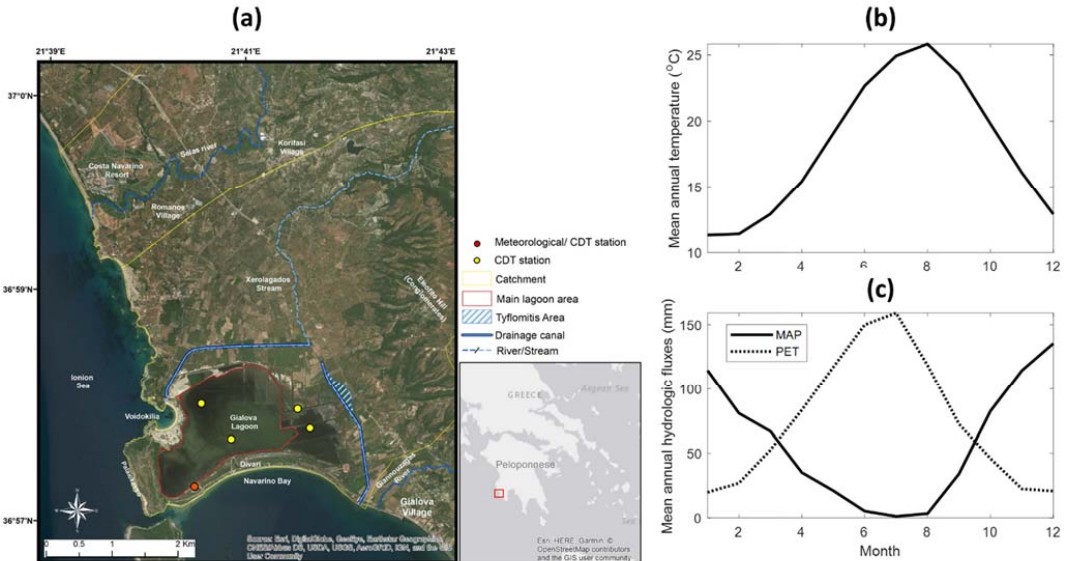

**Fig. 1. a) Map of the study area; b) long-term mean annual precipitation (MAP) and potential evapotranspiration (PET). Sources for panel (a): AeroGRID, CNES/Airbus DS, DigitalGlobe, Earthstar Geographics, Esri, Garmin, GeoEye, GIS user community, HERE, IGN, © OpenStreetMap contributors 2019 (distributed under a Creative Commons BY-SA License), USDA, USGS.**

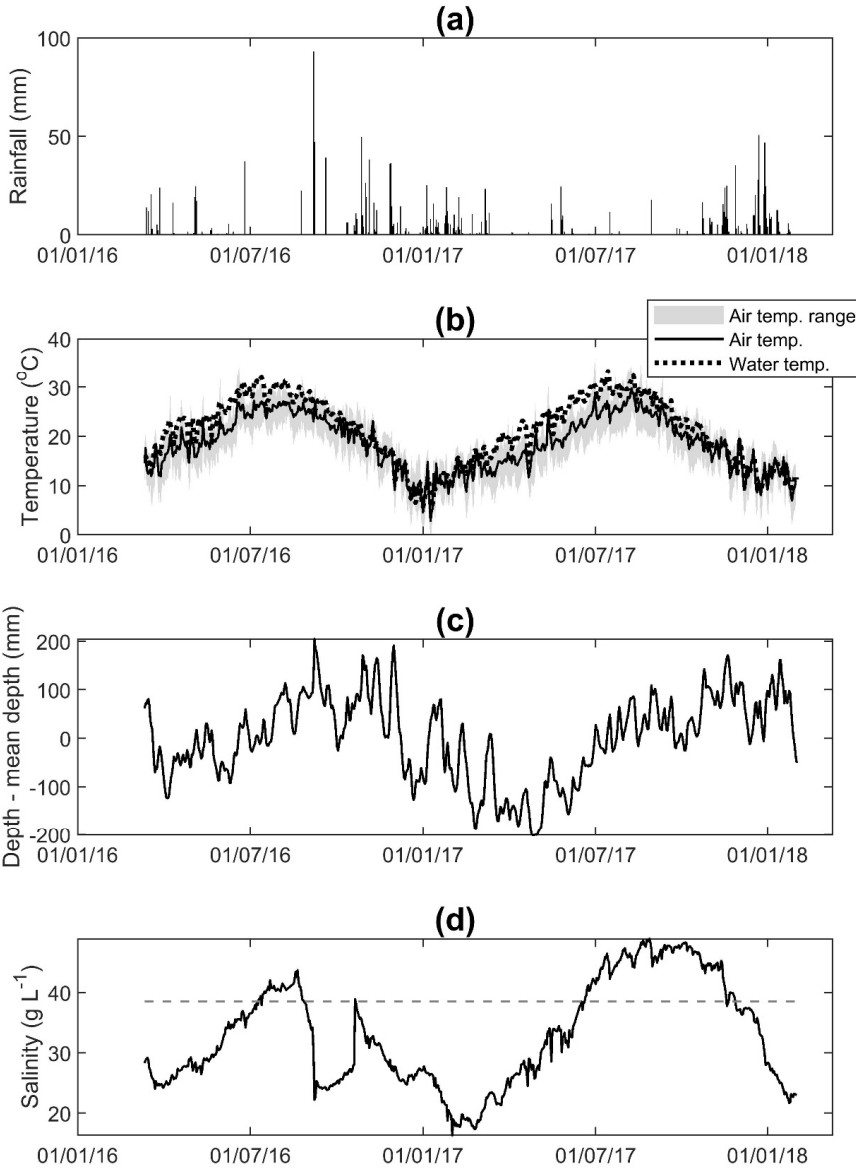


**Fig. 2. Time series of a) daily total precipitation, b) daily mean air and lagoon water temperatures (respectively solid and dashed curves), and daily air temperature range (shaded area), c) lagoon water depth (normalized by the mean depth), and d) salinity.**



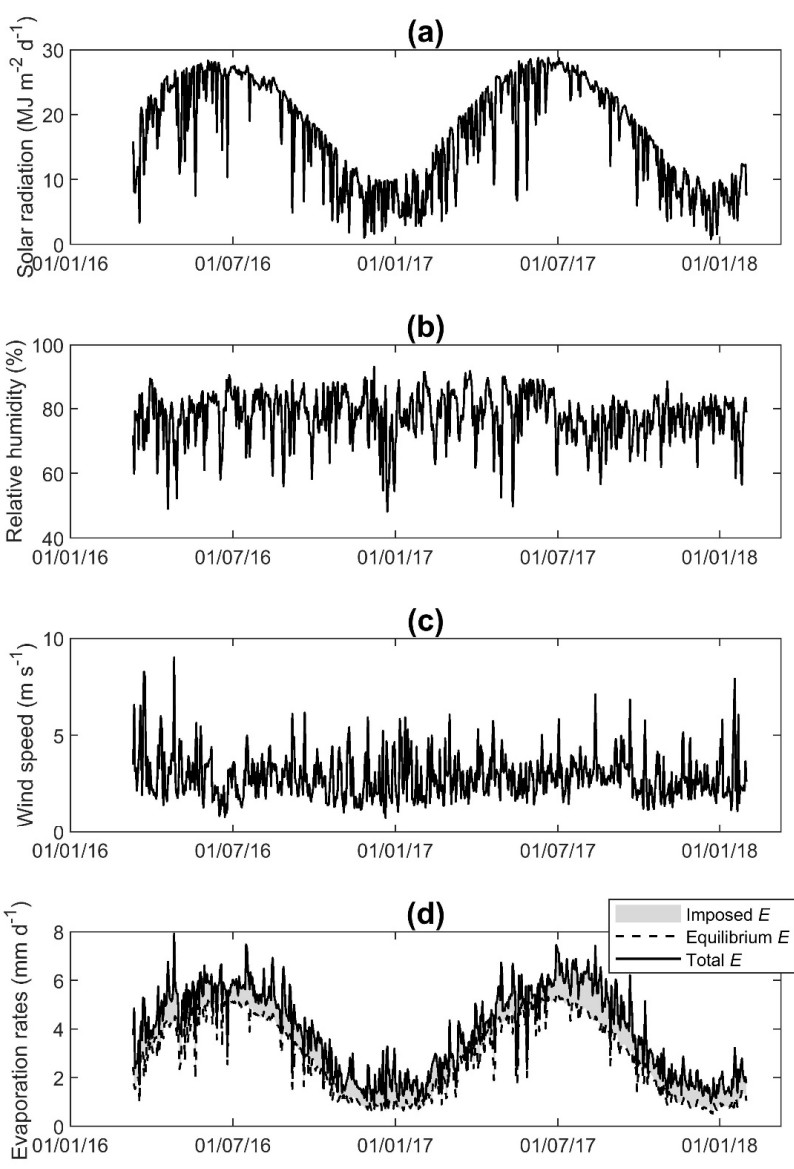

**Fig. 3. Time series of a) daily total incoming shortwave radiation, b) daily mean relative humidity, c) wind speed, and d) calculated evaporation rates (Eq. (7)), further decomposed into equilibrium evaporation (dashed black curve) and 'imposed E'; i.e., the aerodynamic component of evaporation (shaded area).**

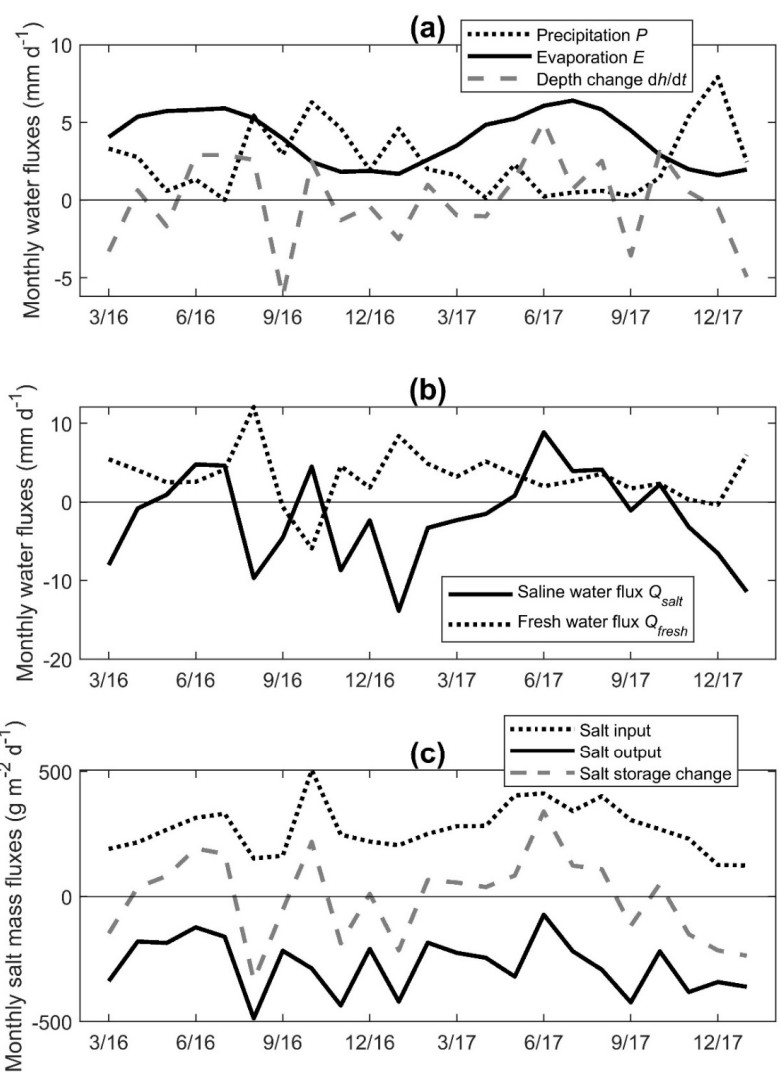

**Fig. 4. a) Mean monthly hydrologic fluxes, b) saline and fresh water exchange fluxes (respectively from equation (10) and (11)), and c) salt mass fluxes. Positive (respectively negative) fluxes indicate water inputs to (outputs from) the Gialova lagoon.**





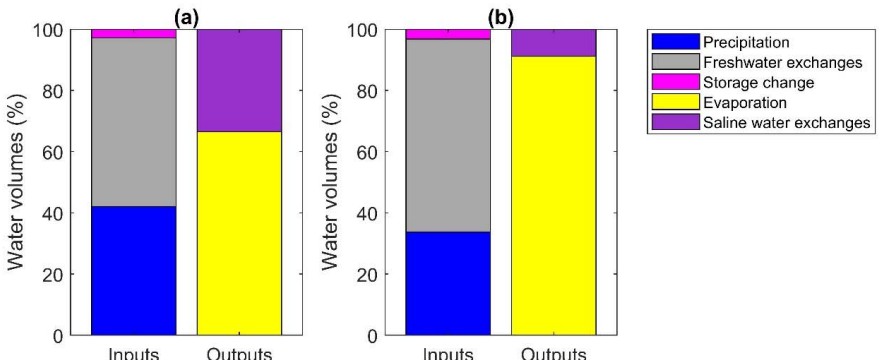

**Fig. 5. Water budgets over the whole study period for a) current conditions and b) future conditions**

**(scenario C3: reduced precipitation and increased evaporation rate, see Table 2). Water volumes are**

**expressed as percentage of the total volume entering or leaving the lagoon.**





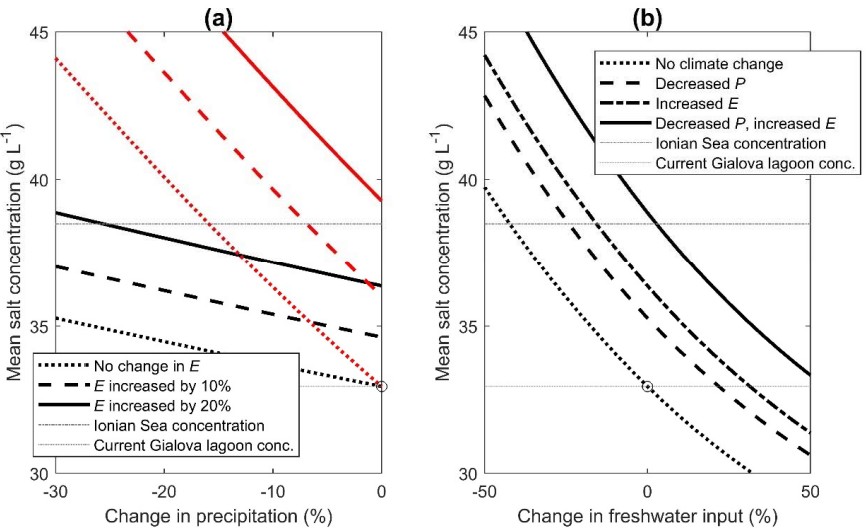

**Fig. 6. a) Effect of climatic changes on the mean salt concentration in the Gialova lagoon (different line**
650 **styles refer to three scenarios for changes in evaporation rate; red lines refer to scenarios where freshwater**
**fluxes are reduced as a result of climatic changes). b) Effect of changes in freshwater input on mean salt**
**concentration, under different climatic scenarios (different line styles; see details in Table 2). For visual**
**reference, the two grey horizontal lines indicate salt concentrations in the Ionian Sea (dot-dashed) and in**
**the Gialova lagoon as of today (dotted); current conditions are indicated by open circles.**




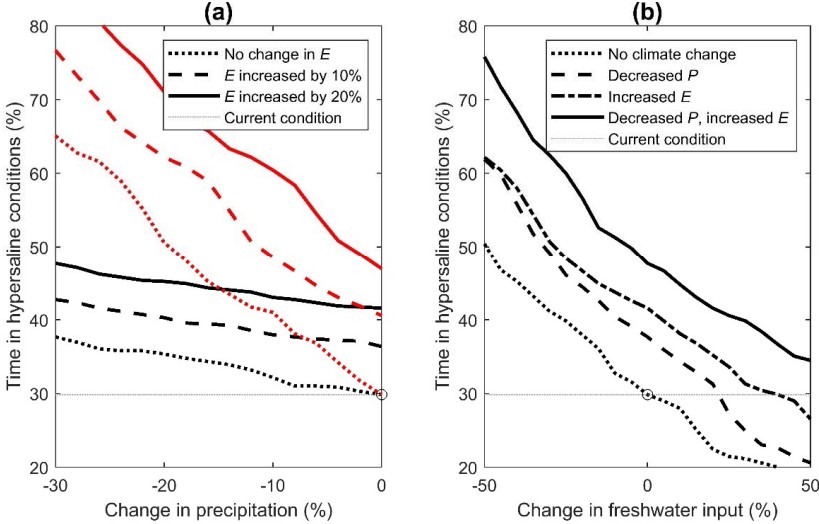

**Fig. 7. a) Effect of climatic changes on the percentage of time under hypersaline conditions (different line styles refer to three scenarios for changes in evaporation rate; red lines refer to scenarios where freshwater fluxes are reduced as a result of climatic changes). b) Effect of changes in freshwater input on the percentage of time under hypersaline conditions, under different climatic scenarios (different line styles; see details in Table 2). Percentage time is calculated for the two simulation years; for visual reference, the grey dotted horizontal lines and the open circles indicate current conditions.**