# Peer review of "Understanding coastal wetland conditions and futures by closing their hydrologic balance: the case of Gialova lagoon, Greece"

_Hydrology and Earth System Sciences, 2019_

## Referee Comment (RC1) · Anonymous Referee #1 · 18 Sep 2019

General comments

The authors investigate salt dynamics in a lagoon, and the effects of climatic changes and water management on salt concentration. They present a clumped model for water and salt mass balance, that when constrained by environmental measurements (precipitation, evaporation) and measurements of water height and concentration, yield estimates of important fluxes between the lagoon and its surroundings (sea, groundwater, etc). The authors use this framework to investigate the fate of the lagoon under future scenarios (increased temperature, less rain), and what would be the required restoration strategies to maintain ecosystem functionality.

The paper is well written and well argued. The methods are sound, and the results are interesting. I enjoyed reading this paper, and I would recommend its publication. Here follow a few comments I would like the authors to address prior to publication.

Specific comments

Lines 125-126. The parenthesis ($C_G$) seems to be misplaced. "the salt mass is obtained as the product of salt concentration ($C_G$) and water depth (h) in the Gialova lagoon."

Line 167-170. This is confusing. You must know the value of h for the first day of measurement, only then you can update h for every time step. From section 2.3.1 we learn that the Northern sensor measures water depth variation, not water depth, but that comes much later in the paper. Please restructure these sentences, convey first that you only measure Delta h/ Delta t, but since you know h from the first day, you can update its value at every time step. Avoid using "absolute value", since it can also mean the function abs(-3)=|-3|=3, it took me a while to realize that "absolute" is used here to contrast "relative", which would be "Delta h".

Lines 175-182. I had to read a few times to understand that once you solve eqs 10 and 11 for the available data, you now use Qfresh (modified), dh/dt, and other data to solve eqs 5 and 6. Ultimately you want $C_G$ from eq 6, and for that you need Qsalt from eq 5. At first I got the impression you were solving a differential equation ("solve in forward mode", line 176), but it seems to be an algebraic relation only. In short, you should restructure this paragraph, it is confusing.

In section 3.2 you first refer to figures 6a and 7a (effects of reduced precipitation), and only then to figures 6b and 7b (managed freshwater). Consider regrouping these 4 panels in the same way you did in the text, figure 6 about the effects of reduced precipitation, and figure 7 about managed freshwater. Consider also labelling the different curves in figures 6 and 7 with C0 through C3 directly on the graphs. If the result is not too cluttered, this would enhance readability.

---

## Author Comment (AC1) · 14 Oct 2019

**Response to Anonymous Referee #1**

We thank Referee #1 for his/her comments, which are addressed as explained below. The Referee's text is reported in Italic and our responses in roman.

*The authors investigate salt dynamics in a lagoon, and the effects of climatic changes and water management on salt concentration. They present a clumped model for water and salt mass balance, that when constrained by environmental measurements (precipitation, evaporation) and measurements of water height and concentration, yield estimates of important fluxes between the lagoon and its surroundings (sea, ground-water, etc). The authors use this framework to investigate the fate of the lagoon under future scenarios (increased temperature, less rain), and what would be the required restoration strategies to maintain ecosystem functionality. The paper is well written and well argued. The methods are sound, and the results are interesting. I enjoyed reading this paper, and I would recommend its publication.*

Thanks for these encouraging comments.

*Here follow a few comments I would like the authors to address prior to publication. Specific comments:*
*Lines 125-126. The parenthesis ($C\_G$) seems to be misplaced. "the salt mass is obtained as the product of salt concentration ($C\_G$) and water depth (h) in the Gialova lagoon."*

The suggested change was implemented.

*Line 167-170. This is confusing. You must know the value of h for the first day of measurement, only then you can update h for every time step. From section 2.3.1 we learn that the Northern sensor measures water depth variation, not water depth, but that comes much later in the paper. Please restructure these sentences, convey first that you only measure Delta h/ Delta t, but since you know h from the first day, you can update its value at every time step.*

Good point. We amended the text by adding the sentence: "This requires setting the initial value of water depth for this recursive calculation to start", just after presenting the recursive equation that updates water depths. In addition, the sentence below Eq. 11 was amended as follows "The two linked Eq. (10) and (11) do not need to be coupled through time because changes in water depth (though not water depth *per se*) and salt concentration in the lagoon are measured"

*Avoid using "absolute value", since it can also mean the function abs(-3)=|-3|=3, it took me a while to realize that "absolute" is used here to contrast "relative", which would be "Delta h".*

The term "absolute" was removed and the sentence now reads "Since the value of water depth *h* varies from one time step to the next, and it is not measured, *h* in Eq. (10) must be updated…"

*Lines 175-182. I had to read a few times to understand that once you solve eqs 10 and 11 for the available data, you now use Qfresh (modified), dh/dt, and other data to solve eqs 5 and 6. Ultimately you want $C\_G$ from eq 6, and for that you need Qsalt from eq 5. At first I got the impression you were solving a differential equation ("solve in forward mode", line 176), but it seems to be an algebraic relation only. In short, you should restructure this paragraph, it is confusing.*

The paragraph has been re-structured as suggested:

"To assess the effects of changing climatic conditions and water resource management on salinity in the Gialova lagoon, we use Eq. (5) and (6) in a forward mode – that is, to estimate salinity variations through time based on hydrologic fluxes. First, Eq. (5) is solved in discretized form to obtain $Q_{salt}$; second, $Q_{salt}$ is inserted in Eq. (6), which is solved also in discretized form to calculate salinity $C_G$. The first step requires estimates of all hydrologic fluxes and the change in water storage except the unknown $Q_{salt}$. Measured precipitation and evaporation rates are modified to account for climatic changes, while the change in storage ($dh/dt$) is maintained from current conditions given the strong coupling of water levels in the lagoon and in the sea (Supplementary materials S1; Fig. S1). Sea level rise is not considered in these scenarios. The $Q_{fresh}$ obtained under current conditions as described in Section 2.4 is modified to account for both climatic and water management changes. With these assumed $P$, $E$, $Q_{fresh}$ and $dh/dt$ values, $Q_{salt}$ is calculated at each time step from the water balance Eq. (5), and salinity is then readily obtained with the salt mass balance Eq. (6)."

*In section 3.2 you first refer to figures 6a and 7a (effects of reduced precipitation), and only then to figures 6b and 7b (managed freshwater). Consider regrouping these 4 panels in the same way you did in the text, figure 6 about the effects of reduced precipitation, and figure 7 about managed freshwater. Consider also labelling the different curves in figures 6 and 7 with C0 through C3 directly on the graphs. If the result is not too cluttered, this would enhance readability.*

Good idea. This rearrangement makes the figure presentation easier. The new figures and captions are reported below. We have also included labels next to the curves of Fig. 7, but not Fig. 6 where curves in some cases include combinations of scenarios, making labelling ambiguous. Finally, we added some details on the imposed variation in precipitation and evaporation in the legend of Fig. 7.

[Figure]

**Fig. 6. Effect of climatic changes on a) the mean salt concentration in the Gialova lagoon and b) the percentage of time under hypersaline conditions. Different line styles refer to three scenarios for changes in evaporation rate; red lines refer to scenarios where freshwater fluxes are reduced as a result of climatic changes. Percentage time is calculated for the two simulation years; for visual reference, the grey horizontal dotted lines indicate current salinity and duration of hypersaline conditions in the Gialova lagoon, and the grey dot-dashed line in a) indicates salt concentration in the Ionian Sea as of today; current conditions are indicated by open circles.**

[Figure]

**Fig. 7. Effect of changes in freshwater input on a) the mean salt concentration in the Gialova lagoon and b) the percentage of time under hypersaline conditions, under different climatic scenarios (different line styles; see details in Table 2). Percentage time is calculated for the two simulation years; for visual reference, the grey horizontal dotted lines indicate current salinity and duration of hypersaline conditions in the Gialova lagoon, and the grey dot-dashed line in a) indicates salt concentration in the Ionian Sea as of today; current conditions are indicated by open circles.**

---

## Referee Comment (RC2) · Anonymous Referee #2 · 24 Mar 2020

Manuscript: Understanding coastal wetland conditions and futures by closing their hydrologic balance: the case of Gialova lagoon, Greece Number: HESSD2019-382 Authors: Stefano Manzoni, Giorgos Maneas, Anna Scaini, Basil E. Psiloglou, Georgia Destouni, Steve W. Lyon

The manuscript presents a coupled water-salt mass balance model for estimating water fluxes of coastal water bodies. The approach was tested for the Gialova lagoon in Greece. Different water sources and fluxes were calculated for today's climate and three different climate change scenarios. The results indicate different and variable contributions of fresh water and salt water fluxes going into and out of the lagoon

throught the year. Future scenarios show that salinity will increase in the lagoon. Therefore, management options have to be developed to maintain freshwater input into the system.

The manuscript is very well written and the results are interesting; also for other coastal wetlands in similar climatic areas. I have some recommendations that are outlined below.

Main comment:

1) What is the salinity of surrounding aquifers? Is the saline water flux solely coming from the Mediterranean Sea or can it also be from groundwater?

2) Evaporation is considered; however, what is the impact of evaporation on the increase of salinity? How is it considered here?

3) The average depth is given (0.6 m). What is the range of depths though? At which depths were samples taken? How is the salt distribution with depths? This can be relevant if salinity increases with depth or if there are local depression within the lagoon with higher salt content that are not completely mixed with the rest of water in the lagoon. Some of it is discussed, but more information could be provided.

4) The water level in the lagoon was kept to current conditions also in the scenarios. How would a decline in groundwater levels due to increased evapo(transpi)ration affect the water levels and the water balance? This could be included in the discussion.

Other, specific comments: - Figure 1a: the map is really small and difficult to identify the lagoon - Figure 1b: in the figure title, it says mean annual precipitation, but actually monthly values are given; error bars would be good to add - Figure S2: error bars (standard deviation) for the spatial average salinity are missing; this applies to other figures showing mean values too; this will help to get some idea about the uncertainty of the calculations - line 326: indicate what hypersaline means in brackets

---

## Author Comment (AC2) · 15 Apr 2020

**Response to Anonymous Referee #2**

We thank Referee #2 for his/her comments, which are addressed as explained below. The Referee's text is reported in Italic and our responses in roman.

*The manuscript presents a coupled water-salt mass balance model for estimating water fluxes of coastal water bodies. The approach was tested for the Gialova lagoon in Greece. Different water sources and fluxes were calculated for today's climate and three different climate change scenarios. The results indicate different and variable contributions of fresh water and salt water fluxes going into and out of the lagoon throught the year. Future scenarios show that salinity will increase in the lagoon. Therefore, management options have to be developed to maintain freshwater input into the system. The manuscript is very well written and the results are interesting, also for other coastal wetlands in similar climatic areas. I have some recommendations that are outlined below.*

Thanks for these encouraging comments and for the suggestions, which were implemented as explained below.

*1) What is the salinity of surrounding aquifers? Is the saline water flux solely coming from the Mediterranean Sea or can it also be from groundwater?*

A network of wells around the Gialova Lagoon is now being monitored (starting after the date of submission of this manuscript) (Pantazis, 2019). Preliminary data show that the aquifers surrounding the Gialova Lagoon are mostly delivering a steady supply of freshwater, but relatively high chloride concentrations have been measured in three wells north of the main body of the lagoon (150-200 mg/l). These wells are in an area of low hydraulic head gradient and thus are prone to inputs of saline water from the lagoon. Most other wells are located on the east side of the lagoon and are representative of conditions in the aquifer that provides freshwater to the Tyflomitis artesian springs.

To include further information (which was not available at the time of submission), we can amend Section 2.1 as follows:

"The aquifers North of the lagoon are prone to salt intrusion from the Gialova lagoon, at least during the dry season (Pantazis, 2019). On the East and South-East sides, wetland areas surround the main lagoon water body. The Tyflomitis artesian springs provide freshwater inputs in this area, thanks to freshwater aquifers feeding them from the East (Pantazis, 2019)."

An additional clarification will also be included in Section 4.2 of the Discussion:

"In that area East of the lagoon, hydraulic gradients are steeper, allowing freshwater aquifers to deliver water to the Tyflomitis springs (Pantazis, 2019)."

*2) Evaporation is considered; however, what is the impact of evaporation on the increase of salinity? How is it considered here?*

Evaporation from the lagoon is an important term in the water balance. By removing water while leaving solutes in the lagoon, any increase in evaporation that is not compensated for by increased precipitation or freshwater inputs will increase salinity. This effect is perhaps best explained by looking at Figure 6, which we plan to modify following the suggestion of

Reviewer #1 (see Response to Anonymous Referee #1). Figure 6a shows that increasing evaporation rate causes salinity to also increase (from dotted to dashed and solid lines). If, in addition to higher evaporation, precipitation and/or freshwater inputs decrease, salinity increases even further. Similar trends are also apparent when looking at the percentage of time the lagoon is expected to be under hypersaline conditions (Figure 6b, originally labelled Figure 7a). The text describing Figure 6a will be amended as follows to highlight the importance of evaporation:

"An increase of 10% in evaporation rate alone (as would be caused by higher temperatures) is predicted to increase salinity by approximately 5%. Any decrease in precipitation in combination with increasing evaporation rate further increases salinity due to accumulation of salt from the marine sources."

Additional explanations of evaporation effects on salinity are provided where climate change scenarios are described. In fact, scenarios C2 and C3 include variations in evaporation rate without or in addition to decreased precipitation, respectively.

*3) The average depth is given (0.6 m). What is the range of depths though?*

The water depth in the Gialova lagoon – based on more than thirty uniformly spaced measurements by Dounas and Koutsoubas (1996) – ranges between 270 and 770 mm (excluding the shoreline and the area immediately adjacent to the channel connecting the lagoon to the sea). In the same study, the overall standard deviation of the water depth is approximately 130 mm. To provide more details on water depth as suggested by the reviewer, we will include the following information at the beginning of Section 2.1:

"The spatial average of water depth in the main water body is approximately 0.6 m, with a range between 0.27 and 0.77 m (Dounas and Koutsoubas, 1996)."

*At which depths were samples taken? How is the salt distribution with depths? This can be relevant if salinity increases with depth or if there are local depression within the lagoon with higher salt content that are not completely mixed with the rest of water in the lagoon. Some of it is discussed, but more information could be provided.*

The continuous measurements presented in this manuscript were taken from probes anchored to solid structures, so they experienced fluctuating water levels. Measurements during the field campaigns were taken at the water surface. The area is generally windy (typically above 2 m/s, see Figure 3 in the manuscript) and the recirculation due to tidal fluctuations is large. For these reasons we regarded the lagoon as vertically well-mixed at the daily time scale. Here we add further evidence in support to this assumption.

We monitored electrical conductivity and temperature at two depths (25 cm depth difference) in the central point of the lagoon. It would be of course ideal to have more vertical profiles, and we will keep this in mind for future work. Unfortunately, equipment malfunctions caused data gaps that do not allow a complete comparison of the conductivity data between the two sensors. However, when both sensors were working, electrical conductivity values were similar. The temperature record from these two sensors is instead nearly complete, and the temperature values at the two depths were highly correlated (Pearson correlation coefficient=0.998; Figure R1). This indicates minimal thermal stratification at the daily time scale and thus well-mixed conditions along the vertical profile.

The result of this analysis will be added in Section 4.1 of the Discussion, where the limitations of our approach are presented:

"Moreover, vertical mixing is ensured by wave motion and tidal fluctuations in this shallow water body. Thermal stratification is minimal at the temporal scale of this study, as indicated by highly correlated water temperatures at the two sampled depths in the central measurement point (Pearson correlation coefficient $r$=0.998)."

[Figure]

**Figure R1. Strong correlation (Pearson correlation coefficient=0.998) between daily mean temperatures measured by two probes at different depths in the central point of the Gialova lagoon. The lower probe was placed 25 cm deeper than the surface probe; both were anchored to a cement pillar and thus experienced fluctuating water levels. The dot-dashed line represents the 1:1 line.**

*4) The water level in the lagoon was kept to current conditions also in the scenarios. How would a decline in groundwater levels due to increased evapo(transpi)ration affect the water levels and the water balance? This could be included in the discussion.*

This is an interesting point we had not considered in the initial submission. Given the strong hydrological coupling between Gialova lagoon and Navarino bay (Supplementary Section S1), we would argue that sea water level will control the water level in the lagoon, regardless of changes in groundwater level. However, lower groundwater could reduce freshwater inputs, altering salinity even if the lagoon water level does not change. In Section 4.2, we will expand the discussion on consequences of catchment-scale hydrologic changes by adding the following sentence along the lines suggested by the reviewer:

"Concurrently, groundwater levels are also expected to decrease, potentially causing saline intrusions around the lagoon – a process already ongoing at least at the end of the dry season (Pantazis, 2019)."

In Section 4.3, we will also remind the reader that groundwater will be impacted by higher water demand in the catchment:

"These estimates do not reflect the expected reduction in precipitation and increase evaporation in the coming decades, which will decrease both runoff to and groundwater flow in the lagoon."

*Other, specific comments:*
*- Figure 1a: the map is really small and difficult to identify the lagoon; Figure 1b: in the figure title, it says mean annual precipitation, but actually monthly values are given; error bars would be good to add*

We agree that Figure 1a can be improved along these lines: the area around the Gialova lagoon will be enlarged and font size will be increased. In Figure 1b-c, we can include a shaded band around the mean to indicate the variability. Since these hydrological variables are not normally distributed, it is more instructive to show the variability between the 5th and 95th percentiles rather than the standard deviation. The revised Figure 1b-c will be described with these additional sentences in Section 2.1:

"Seasonal patterns are clear in all these hydro-climatic variables, with dry and hot summers and wet and mild winters. Precipitation shows higher variability than temperature and potential evapotranspiration, as indicated by larger differences between 5th and 95th percentiles (shaded areas in Fig. 1b-c)."

[Figure]

**Fig. 1b-c. b) long-term mean annual temperature; c) long-term mean annual precipitation (MAP) and potential evapotranspiration (PET). In panels (b) and (c), shaded areas indicate the variability around the mean (5th and 95th percentiles).**

*- Figure S2: error bars (standard deviation) for the spatial average salinity are missing; this applies to other figures showing mean values too; this will help to get some idea about the uncertainty of the calculations*

In Figures S2 and S4, values are areal averages, that is, weighed averages based on the specific sampling design of each field campaign. While the design was generally comparable (sampling points along the shoreline and some in the central part of the lagoon), the number of sampling points differed throughout the years, so standard deviations and standard errors will also vary because of these differences, making them difficult to interpret. For this reason, we would prefer keeping Figures S2 and S4 as they are in the initial submission.

To give an idea of the degree of spatial variability and its consistency through time, we calculated the standard deviations of salinity from the study by Dounas and Koutsoubas (1996). Data refer to the 1995-1996 field campaign and are shown in aggregated form in Figures S2 and S4. Four measurements uniformly spread in the lagoon at each of 22 sampling dates were considered to calculate the standard deviation, resulting in an average standard deviation of 4.6 g/l (range: 0.8 to 12.2 g/l; 1st and 3rd quartiles: 1.9 and 6.9 g/l, respectively). The seasonal variations in salinity are in the order of 30 g/l (Figure 2), suggesting that spatial variability is relatively low compared to temporal variability, consistent with the model assumptions.

*- line 326: indicate what hypersaline means in brackets.*

For clarity, we will amend the sentence where the term "hypersaline" first appears as follows:

"Salinity tends to increase during the spring and throughout the summer, peaking in late summer or early fall (hypersaline conditions, defined here with respect to the average seawater salinity of 38.5 g L$^{-1}$)."

**References**

Dounas K. and D. Koutsoubas (1996). Environmental Impact Study on pollution from petroleum products in Navarino Bay and Gialova Lagoon wetland, Institute of Marine Biology of Crete, 298 pages (available in Greek).

Pantazis C. (2019). Ecosystem services and groundwater quality: the case study of Gialova Lagoon, MSc thesis, National Technical University of Athens, Metsovion Interdisciplinary Research Center, 71 pages.

---

## Editor Comment (EC1) · Dimitri Solomatine (Editor) · 20 Apr 2020

The evaluation by reviewers is overall quite positive. They provided useful comemnts and clearly formulated recommendations. authors replies show, that these comments are understood and they have a clear plan how ĐšÑĹĐřĐřÑĹÑĄĐşĐťĐţ ĐţÑĹÑÑČÑÑÑŐÑŐÑŐthe manuscript will be revised. They are invited to do so. I wish theauthors good luck in doing this, during these difficult times...

---

## Author Response (AR1)

**Response to the Editor's Comments**

We would like to thank the editor for offering the opportunity to submit a revised manuscript following the suggestions of the reviewers. We have already detailed how we planned to revise our manuscript in the responses to the reviewers' comments posted in the public discussion forum. In this letter, we only describe how the planned changes have been implemented, and leave our arguments in support to those changes in the open discussion to avoid unnecessary repetitions. The differences between planned and actual changes are mostly editorial, but we also improved the Discussion by providing context for our findings in a more quantitative way (comparison of water input and output partitioning among three Mediterranean lagoons; L456-475). Finally, the manuscript was proof-read and other minor editorial changes were made throughout the text. Line and figure numbers refer to the revised manuscript.

**Response to Anonymous Referee #1**

We thank Referee #1 for his/her comments, which are addressed as explained below. The Referee's text is reported in Italic and our responses in roman.

*The authors investigate salt dynamics in a lagoon, and the effects of climatic changes and water management on salt concentration. They present a clumped model for water and salt mass balance, that when constrained by environmental measurements (precipitation, evaporation) and measurements of water height and concentration, yield estimates of important fluxes between the lagoon and its surroundings (sea, ground-water, etc). The authors use this framework to investigate the fate of the lagoon under future scenarios (increased temperature, less rain), and what would be the required restoration strategies to maintain ecosystem functionality. The paper is well written and well argued. The methods are sound, and the results are interesting. I enjoyed reading this paper, and I would recommend its publication.*

Thanks for these encouraging comments.

*Here follow a few comments I would like the authors to address prior to publication. Specific comments:*
*Lines 125-126. The parenthesis (C_G) seems to be misplaced. "the salt mass is obtained as the product of salt concentration (C_G) and water depth (h) in the Gialova lagoon."*

The text was amended as follows:
L144: "In Eq. (3), the salt mass is obtained as the product of salt concentration ($C_G$) and water depth (h) in the Gialova lagoon."

*Line 167-170. This is confusing. You must know the value of h for the first day of measurement, only then you can update h for every time step. From section 2.3.1 we learn that the Northern sensor measures water depth variation, not water depth, but that comes much later in the paper. Please restructure these sentences, convey first that you only measure Delta h/ Delta t, but since you know h from the first day, you can update its value at every time step.*

The text was amended as follows:
L190: "The two linked Eq. (10) and (11) do not need to be coupled through time because changes in water depth (though not water depth *per se*) and salt concentration in the lagoon are measured"

L194: "This requires setting the initial value of water depth for this recursive calculation to start"

*Avoid using "absolute value", since it can also mean the function abs(-3)=|-3|=3, it took me a while to realize that "absolute" is used here to contrast "relative", which would be "Delta h".*

The term "absolute" was removed and the sentence now reads:
L192: "Since the value of water depth h varies from one time step to the next, and it is not measured, h in Eq. (10) must be updated…"

*Lines 175-182. I had to read a few times to understand that once you solve eqs 10 and 11 for the available data, you now use Qfresh (modified), dh/dt, and other data to solve eqs 5 and 6. Ultimately you want C_G from eq 6, and for that you need Qsalt from eq 5. At first I got the impression you were solving a differential equation ("solve in forward mode", line 176), but it seems to be an algebraic relation only. In short, you should restructure this paragraph, it is confusing.*

The paragraph has been re-structured as suggested:
L201: "To assess the effects of changing climatic conditions and water resource management on salinity in the Gialova lagoon, we use Eq. (5) and (6) in a forward mode – that is, to estimate salinity variations through time based on hydrologic fluxes. First, Eq. (5) is solved in discretized form to obtain $Q_{salt}$; second, $Q_{salt}$ is inserted in Eq. (6), which is solved also in discretized form to calculate salinity $C_G$. The first step requires estimates of all hydrologic fluxes and the change in water storage except the unknown $Q_{salt}$. Measured precipitation and evaporation rates are modified to account for climatic changes, while the change in storage ($dh/dt$) is maintained from current conditions given the strong coupling of water levels in the lagoon and in the sea (Supplementary materials S1; Fig. S1). Sea level rise is not considered in these scenarios. The $Q_{fresh}$ obtained under current conditions as described in Section 2.4 is modified to account for both climatic and water management changes. With these assumed $P$, $E$, $Q_{fresh}$ and $dh/dt$ values, $Q_{salt}$ is calculated at each time step from the water balance Eq. (5), and salinity is then readily obtained with the salt mass balance Eq. (6)."

*In section 3.2 you first refer to figures 6a and 7a (effects of reduced precipitation), and only then to figures 6b and 7b (managed freshwater). Consider regrouping these 4 panels in the same way you did in the text, figure 6 about the effects of reduced precipitation, and figure 7 about managed freshwater. Consider also labelling the different curves in figures 6 and 7 with C0 through C3 directly on the graphs. If the result is not too cluttered, this would enhance readability.*

We re-arranged the panels of Figures 6 and 7 as suggested, and updated the captions accordingly. We have also included labels next to the curves of Fig. 7.

**Response to Anonymous Referee #2**

We thank Referee #2 for his/her comments, which are addressed as explained below. The Referee's text is reported in Italic and our responses in roman.

*The manuscript presents a coupled water-salt mass balance model for estimating water fluxes of coastal water bodies. The approach was tested for the Gialova lagoon in Greece. Different water sources and fluxes were calculated for today's climate and three different climate change scenarios. The results indicate different and variable contributions of fresh water and salt water fluxes going into and out of the lagoon throught the year. Future scenarios show that salinity will increase in the lagoon. Therefore, management options have to be developed to maintain freshwater input into the system. The manuscript is very well written and the results are interesting, also for other coastal wetlands in similar climatic areas. I have some recommendations that are outlined below.*

Thanks for these encouraging comments and for the suggestions, which were implemented as explained below.

*1) What is the salinity of surrounding aquifers? Is the saline water flux solely coming from the Mediterranean Sea or can it also be from groundwater?*

The text was amended as follows:
L83: "On the East and South-East sides, wetland areas surround the main lagoon water body. The Tyflomitis artesian springs provide freshwater to this area, thanks to freshwater aquifers feeding them from the East (Pantazis, 2019)."
L90: "However, this aquifer is prone to salt intrusion from the Gialova lagoon during the dry season (Pantazis, 2019)."
L480: "In that area East of the lagoon, hydraulic gradients are steeper, allowing freshwater aquifers to deliver water to the Tyflomitis springs (Pantazis, 2019)."

*2) Evaporation is considered; however, what is the impact of evaporation on the increase of salinity? How is it considered here?*

The text was amended as follows:
L366: "An increase of 10% in evaporation rate alone (as would be caused by higher temperatures) is predicted to increase salinity by approximately 5%. Any decrease in precipitation in combination with increasing evaporation rate further increases salinity due to accumulation of salt from the marine sources."
L371: "However, for a given evaporation rate, when both precipitation and freshwater inputs are decreased, salinity increases more than when only precipitation is decreased (compare red and black lines)."

*3) The average depth is given (0.6 m). What is the range of depths though?*

The following information was added:
L73: "The spatial average of water depth in the main water body is approximately 0.6 m, with a range between 0.27 and 0.77 m (Dounas and Koutsoubas, 1996)."

*At which depths were samples taken? How is the salt distribution with depths? This can be relevant if salinity increases with depth or if there are local depression within the lagoon with*

*higher salt content that are not completely mixed with the rest of water in the lagoon. Some of it is discussed, but more information could be provided.*

The following text was added:
L411: "Moreover, vertical mixing is ensured by wave motion and tidal fluctuations in this shallow water body. Thermal stratification is minimal at the temporal scale of this study, as indicated by highly correlated water temperatures at the two sampled depths in the central measurement point (Pearson correlation coefficient $r$=0.998)."

*4) The water level in the lagoon was kept to current conditions also in the scenarios. How would a decline in groundwater levels due to increased evapo(transpi)ration affect the water levels and the water balance? This could be included in the discussion.*

The following text was added:
L513: "Concurrently, groundwater levels are also expected to decrease, potentially causing saline intrusions around the lagoon – a process already ongoing at least at the end of the dry season (Pantazis, 2019)."
L572: "These estimates do not reflect the expected reduction in precipitation and increase evaporation in the coming decades, which will decrease both runoff to and groundwater flow in the lagoon."

*Other, specific comments:*
*- Figure 1a: the map is really small and difficult to identify the lagoon; Figure 1b: in the figure title, it says mean annual precipitation, but actually monthly values are given; error bars would be good to add*

Figure 1 was improved as suggested and the revised Figure 1b-c is described with these additional sentences:
L97: "Seasonal patterns are clear in all these hydro-climatic variables, with dry and hot summers and wet and mild winters. Precipitation shows higher variability than temperature and potential evapotranspiration, as indicated by larger differences between the 5th and 95th percentiles of the monthly values (shaded areas in Fig. 1b-c)."

*- Figure S2: error bars (standard deviation) for the spatial average salinity are missing; this applies to other figures showing mean values too; this will help to get some idea about the uncertainty of the calculations*

No specific changes were made, but please see our response for a complete motivation.

*- line 326: indicate what hypersaline means in brackets.*

The text was amended as follows:

[revised manuscript text omitted]